# Sensory dynamics of visual hallucinations in the normal population

**Joel Pearson[1]\*, Rocco Chiou[1], Sebastian Rogers[1], Marcus Wicken[1], Stewart Heitmann[2], Bard Ermentrout[2]**

[1]The School of Psychology, University of New South Wales, Sydney, Australia; [2]Department of Mathematics, University of Pittsburgh, Pittsburgh, United States

**Abstract** Hallucinations occur in both normal and clinical populations. Due to their unpredictability and complexity, the mechanisms underlying hallucinations remain largely untested. Here we show that visual hallucinations can be induced in the normal population by visual flicker, limited to an annulus that constricts content complexity to simple moving grey blobs, allowing objective mechanistic investigation. Hallucination strength peaked at ~11 Hz flicker and was dependent on cortical processing. Hallucinated motion speed increased with flicker rate, when mapped onto visual cortex it was independent of eccentricity, underwent local sensory adaptation and showed the same bistable and mnemonic dynamics as sensory perception. A neural field model with motion selectivity provides a mechanism for both hallucinations and perception. Our results demonstrate that hallucinations can be studied objectively, and they share multiple mechanisms with sensory perception. We anticipate that this assay will be critical to test theories of human consciousness and clinical models of hallucination.

## Introduction

Hallucinations occur across a wide range of pathologies and are also common in non-clinical populations (*Barrett, 1993*; *1994*). Generally, hallucinations are defined as an involuntary percept-like experience in the absence of an appropriate direct stimulus (*Bentall, 1990*). However, very little is known about the mechanisms underlying hallucinations, due largely to the methodological constraints caused by their inherently subjective, constantly changing heterogeneous content.

Visual hallucinations are thought to arise in exceptional circumstances when external stimuli are overwhelmed by internally generated spontaneous patterns of neural activity. This situation occurs when the parameters governing normal visual function are altered due to changes in brain anatomy or physiology (*Ffytche, 2008*; *Butler et al., 2012*), state changes such as dreaming or migraines (*Llinás and Ribary, 1993*; *Aurora and Wilkinson, 2007*), psychotropic drugs that temporarily perturb normal cortical function, or empty full field luminance flicker (*Passie et al., 2008*; *Billock and Tsou, 2012*). However, across these classes of hallucination, understanding has been severely limited by the multi-feature (color, form and motion) heterogeneous content that changes unpredictably over time, and typically requires subjective reports or subsequent depiction such as drawing to communicate subjective experience (*Allefeld et al., 2011*). To study visual hallucinations, we constrained empty-field flicker to a thin annulus that was centred on the fovea (*Figure 1A*) . This stimulus effectively constrained the hallucinated forms to one spatial dimension.

## Results

When a white annulus was flickered on/off on a black background (~2–30 Hz), we noticed that light grey blobs appeared and rotated around the annulus, first in one direction then the other

\*For correspondence: jpearson@ unsw.edu.au

**Competing interests:** The authors declare that no competing interests exist.

**eLife digest** Hallucinations can occur in both healthy and unwell people. Drugs, sleep deprivation, loss of vision, and migraines can all trigger visual hallucinations in people with no psychiatric illness. We have known for more than 200 years that flickering light can induce hallucinations in almost anyone. However, the unpredictability, complexity and personal nature of hallucinations make them difficult to measure scientifically, and previous studies often had to rely on drawings and verbal descriptions.

Pearson et al. now show how to induce visual hallucinations in anyone , and how to measure them objectively and reliably without relying on subjective reports or drawings. The participant volunteers were university students with no history of migraines or psychiatric disorders. The students watched an image of a plain white ring flicker on and off around 10 times per second against a black background. All individuals reported seeing pale grey blobs appear in the ring and rotate around it, first in one direction and then the other. These grey hallucinations are much simpler than the complex multi-shape hallucinations people generally experience and so they are easier to study objectively.

To measure the hallucinations, Pearson et al. placed a second ring marked with permanent perceptual grey blobs inside the white ring. By stating whether the hallucinated blobs were lighter or darker than the real blobs, the participants were able to communicate the strength of their hallucinations. Similarly, by indicating when the hallucinated blobs had moved past fixed lines at the top and bottom of the white ring, the subjects were able to convey the speed of the hallucinated motion.

The hallucinated blobs and 'real' perceived blobs had many of the same properties, and seemed to arise in the same part of the brain, the visual cortex. By using the data to construct a neural computer model of visual cortex, Pearson et al. propose a mechanism that can explain both normal vision and hallucinations. The next step is to investigate whether the experimental methods can also model the hallucinations produced by psychiatric disorders.

(*Figure 1A*; see *Videos 1* and *2*). Unlike full field luminance flicker stimulation whose content changes as a function of flicker frequency (*Allefeld et al., 2011*; *Mauro et al., 2015*), the light grey blobs remained clearly observable across the range of oscillation frequencies tested. This reduces the visual feature dimensions and overcomes many of the prior limitations set by the multi-feature heterogeneous content in pathological, spontaneous and full field flicker induced hallucinations.

To measure the strength of this hallucination, we devised a technique allowing us to assess the effective contrast with a two alternative forced choice procedure. We presented an interior annulus housing physical sinusoidal luminance modulation simultaneously with the flickering annulus (*Figure 1B*). We presented this physical retina-sourced annulus at a range of different contrasts and participants reported whether it was higher or lower in contrast than any content in the flickering empty annulus. Across two experiments, data from 28 and 24 subjects were fit with cumulative Gaussian functions to give an estimate of the point of subjective contrast equivalence. This gave us a proxy of the effective contrast of the hallucinated blob structures across different flicker frequencies. *Figure 1D* shows the effective contrast of the hallucinations as a function of flicker frequency. Contrast estimates peaked around 11 Hz, then continued to decline slowly as a function of frequency (main effect frequency: exp. 1A $F(3,81) = 3.42$, p=0.021; exp. 1B: $F(3,69) = 5.81$, p=0.001).

One proposition is that flicker induced hallucinations might be largely due to an interaction between perception and retinal after effects (*Bidwell, 1897*). To test for a cortical contribution to the hallucinated blobs, we devised a version of the hallucination-contrast experiment that depended on cortical interocular cross-talk. We presented two small annuli, one to each eye using a mirror stereoscope, and flickered both rings either synchronously or asynchronously at 2.5 Hz to 20 new participants. In the asynchronous condition binocular neurons should receive flicker stimulation at ~5 Hz (rather than 2.5 Hz). Accordingly, if the hallucinated content is the product of binocular neurons, viewers should experience higher contrast in the asynchronous condition as 5 Hz is closer to the 11 Hz peak in contrast, we found in the first experiment. *Figure 1E* shows exactly this; hallucinations

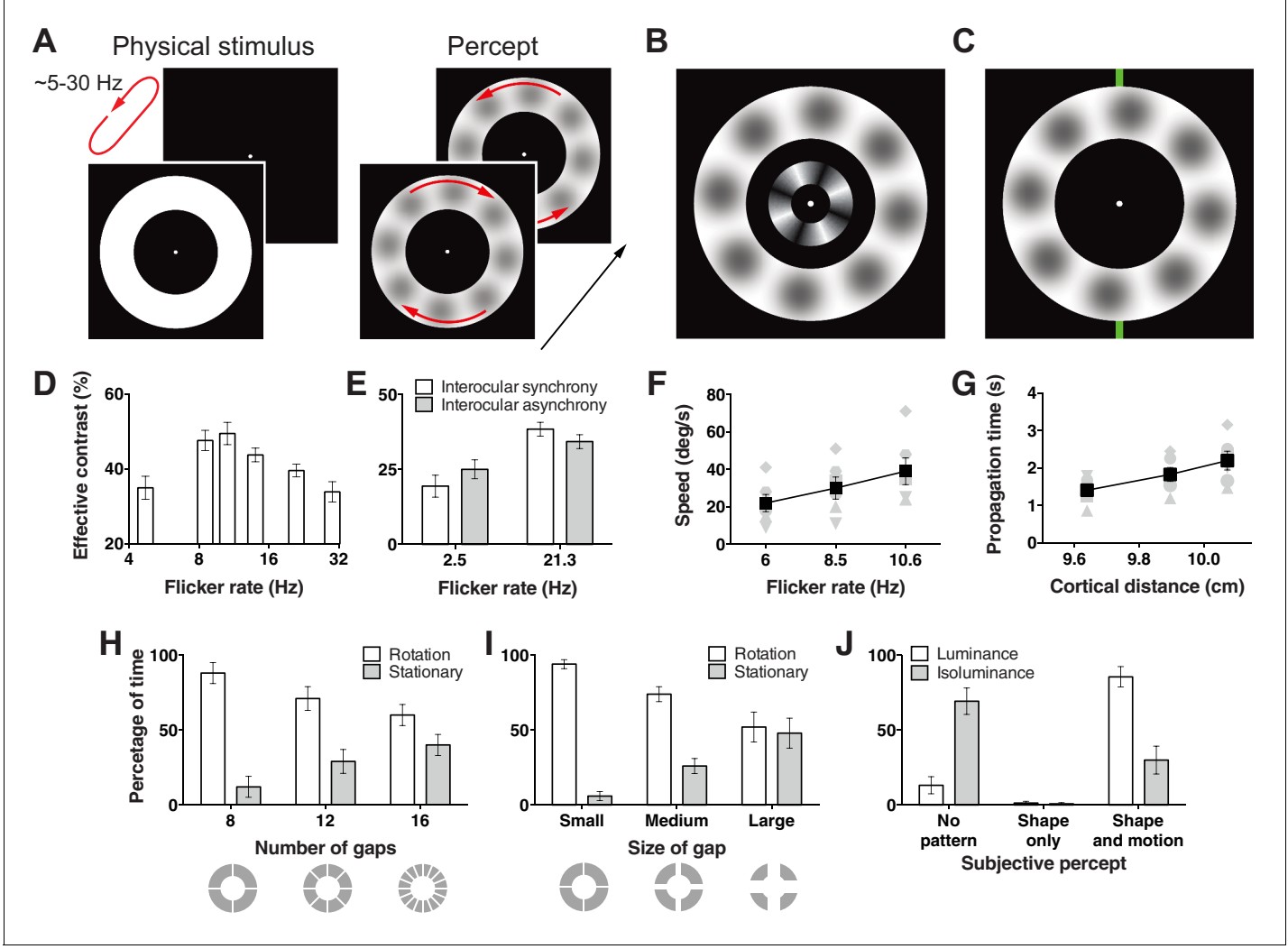

**Figure 1.** Hallucination stimulus and data. (A) Physical stimulus and depiction of the percept. (B) Depiction of the stimulus used to measure the effective contrast. The small inner annulus is the perceptual, while the larger outer annulus shows a depiction of the hallucinated content. (C) Hallucination depiction and nonius lines used to measure the effective rotation propagation times. (D) Effective contrast data showing mean point of subjective equivalence between the perceptual and hallucinated content as a function of flicker frequency (Exp. 1; N = 42, 56 trials per frequency). (E) Data showing interocular interaction (Exp. 2; N = 20, 56 trials per combination of synchrony and frequency). Synchronous and asynchronous flickering annuli give different contrasts measures. (F) Hallucination motion speed measures using stimulus in C, as a function of flicker rate (Exp. 3A; N = 6, 40 trials per frequency). (G) Dependence of propagation times on cortical distance (Exp. 3B; N = 6, 40 trials per eccentricity condition). Distance around the annulus was converted into cm across cortex using the formulae from (**Horton and Hoyt, 1991**). Main effect of distance $F(2, 5) = 13.74$; p=0.001. (H) The effect of number of physical gaps in the annulus (Exp. 4A; N = 4, 10 trials per stimulus). (I) The effect of the width of physical gaps in the annulus (Exp. 4B; N = 4, 10 trials per stimulus). (J) Hallucinated structure and motion is reduced at isoluminance (Exp. 5; N = 5, 20 trials per luminance condition). All error bars show ± SEM.

in the asynchronous 2.5 Hz condition were perceived at a higher contrast than in the synchronous condition ($t(19) = 2.60$, p=0.018). Likewise, the same experiment performed at 21 Hz yielded the opposite pattern of results, the synchronous condition gave higher contrast values ($t(19) = 2.28$, p=0.034; interaction: $F(1, 19) = 8.06$, p=0.011), as hallucinations produced by >21 Hz flicker were perceived as lower in contrast in our first experiment (**Figure 1D**). Together these data suggest that flicker induced hallucinations transpire at or beyond binocular neurons and cannot be the sole product of retinal afterimages.

To measure the motion dynamics of rotation in this hallucination, we devised a technique allowing us to quantify the speed of rotation similar to that used in (**Wilson et al., 2001**). Observers

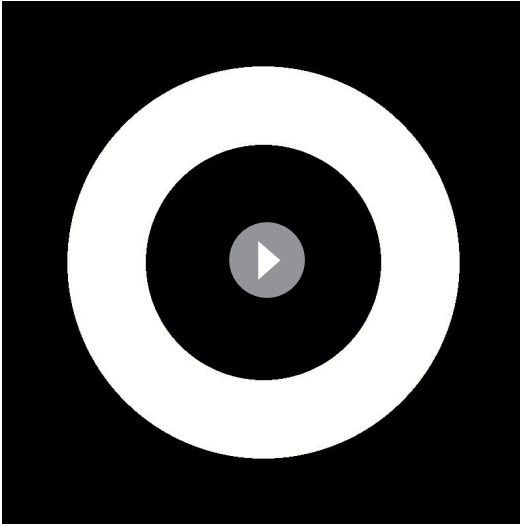

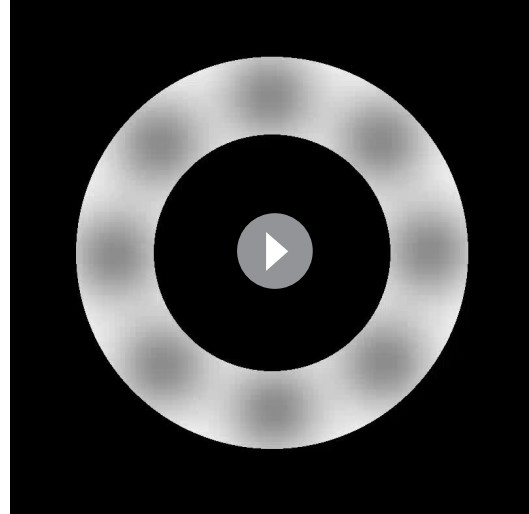

**Video 1.** An animated movie representation of one of our stimuli. Under the right viewing conditions, you may experience light grey blobs (that are not physically presented in the movie) appearing around the flickering annulus. This video contains flashing and alternating images, and therefore might not be suitable for readers with photosensitive epilepsy.

**Video 2.** A perceptual, retina-based representation approximating what some see in the otherwise empty flickering white annulus of *Video 1*. Note, this is only one interpretation, the individual hallucination experience may vary from individual to individual. This video contains flashing and alternating images, and therefore might not be suitable for readers with photosensitive epilepsy.

depressed and held a key when a monitored section of the rotating hallucinatory pattern passed a nonius line at the top of the annulus (*Figure 1C*). Observers then released the key only when the monitored section of the pattern reached a bottom nonius line, marking a travelled distance of half a rotation. This technique gave us propagation times for a fixed annulus distance, hence the speed of the travelling hallucination. The rotational speed was dependent on the annulus flicker rate (*Figure 1F*), with higher frequencies giving faster rotation speeds ($F_{(2,10)} = 24.75$, p<0.001).

To learn how rotation speed varies with eccentricity, we scaled our entire stimulus across three different mean annular radii and using the same timing procedure mapped speed in 6 observers. Using the same speed procedure as above, the mean speed for eccentricities of 6.19°, 7.96°, and 9.67° (mean radii) were 34, 33 and 33° s$^{-1}$. Based on the hypothesis that the hallucinated structure and motion originates in primary visual cortex we converted visual distance into physical distance on the cortex in cm using the detailed surface map of human V1 (*Horton and Hoyt, 1991*). *Figure 1G* shows longer propagation times as a function of greater neural distance, corresponding to a mean propagation speed of 5.6 cm s$^{-1}$ over V1 surface.

Next we tested whether the hallucinations could propagate across gaps in the flickering annulus stimulus. We added 4, 8, and 12 permanent gaps (0.59° width) into the flickering annulus at cardinal locations and sub-cardinal divisions (*Figure 1H*). Eight observers monitored the direction of motion of the hallucinated blobs by holding one of the designated keys down or reported no clear direction by releasing keys for multiple 60-second durations. *Figure 1H* shows the percentage of time observers reported rotational motion vs. stationary patterns as a function of gap number. There was a clear trend of less rotation with a greater number of gaps ($F_{(2,6)} = 14.68$, p=0.005). Next we held the number of gaps constant at four, and manipulated gap size across three different values (0.59°, 1.78°, 2.97°), while observers again tracked hallucinatory motion or its absence. Again, the percentage of time rotational motion was perceived went down as a function of gap size ($F_{(2,6)} = 15.93$, p=0.004) (*Figure 1I*).

To learn if these hallucinations are specific to luminance flicker or generalize to isoluminant color flicker, we ran a new experiment with five observers who tracked static shapes, moving shapes, or no structure at all, for both the standard luminance flicker and subjective isoluminant red and green

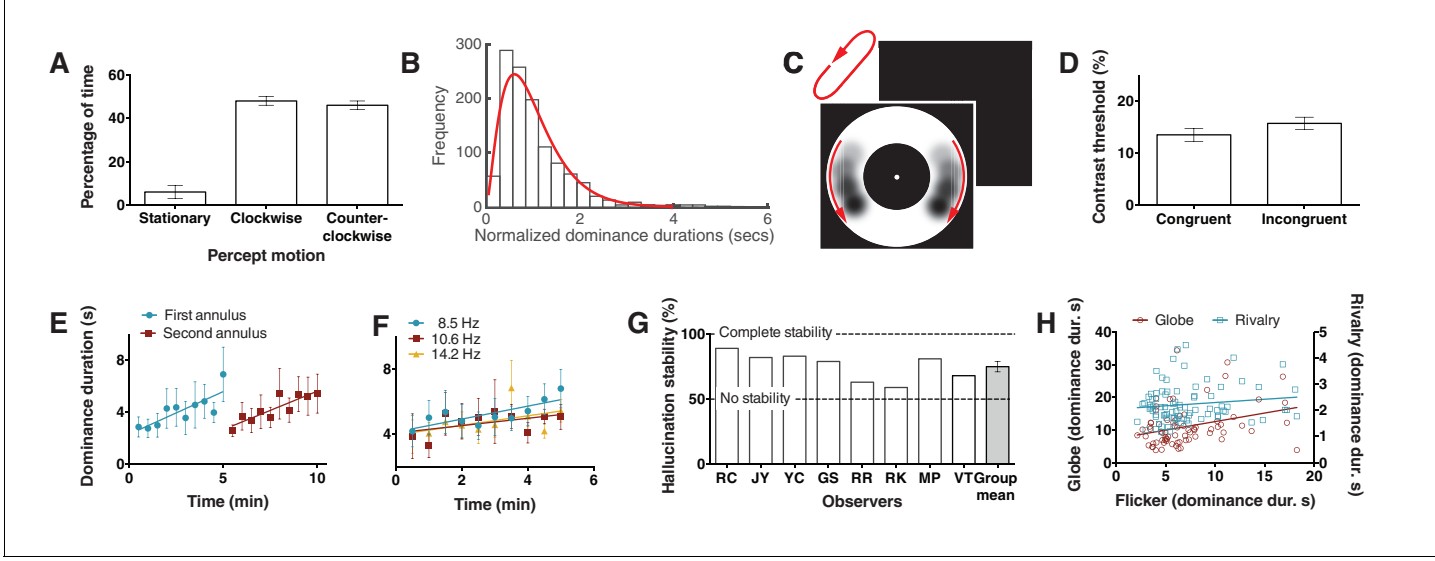

**Figure 2.** Bistability of the hallucination. (**A**) Data showing that subjects experience the hallucination rotation equally in clockwise and counter-clockwise directions (Exp. 6; *N* = 9, 10 min of tracking). (**B**) A histogram of dominance durations, showing the long tail and fit with a gamma function, a hallmark of perceptual bistability (Exp. 6; from the gamma fit: a = 2.49; b = 0.40). (**C**) Depiction of the motion probe stimulus. Motion probes were presented moving clockwise or counter-clockwise, while subjects tracked hallucination alternations. (**D**) Contrast thresholds from the probe stimulus for congruent and incongruent probes (Exp. 7; *N* = 13, 384 trials). (**E**) Longer dominance durations over viewing time suggest a form of adaptation that is local in visual space, the second ring 'resets' the adaptation (Exp. 8A; *N* = 7, 10 min tracking per stimulus order). (**F**) Dominance durations over time for three different flicker frequencies (Exp. 8B; *N* = 6, 10 trials per frequency). (**G**) 8 individual subjects and the mean stability for intermittent physical presentations (Exp. 9; 100 trials). (**H**) Comparing dominance durations in binocular rivalry, rotation globe-stimulus and flicker hallucinations (Exp. 10; *N* = 84, 2 trials of two minutes tracking). Retest reliability: flicker: *r* =0.736; globe: *r* =0.741; rivalry: *r* =0.744; all *p*s <0.0001. All error bars show ± SEM.

annuli. We first assessed subjective isoluminance in each observer using the flicker fusion technique (*Wagner and Boynton, 1972*). Using these color values we flickered annuli at 6 or 12 Hz for 30 s. *Figure 1J* shows the percentage of time reported for each category (no shape, shape only, and shape + motion) for luminance and isoluminance stimuli collapsed across flicker frequency (main effect and all interactions for flicker frequency: all *p*s > 0.29). Strikingly, unlike luminance stimuli, observers reported no pattern for over 60% of the time at isoluminance (*F*(2,8) = 39.67, p<0.001). These data suggest a neural locus sensitive to luminance, but not color flicker, likely the dorsal visual processing stream, which has a higher proportion of cells blind to isoluminance (*Shapley, 1990*). However, we only tested isoluminant red/green, so it remains unknown if this pattern would extend to blue/yellow stimuli.

Over the past several decades researchers have been fascinated by perceptual bistability as a method to study the neural correlates of consciousness and how the brain makes decisions (*Blake and Logothetis, 2002*). Accordingly, we wondered if the motion in this hallucination might indeed be bistable and hence open a new window into the study of how the brain makes choices for conscious experience. First, to investigate if people hallucinated clockwise and counter-clockwise motion equal proportions of time, 9 observers tracked rotation direction for 10 min. *Figure 2A* shows equivalent percentages of time reported for each motion direction (CW vs CCW: *t*(8) < 1, p=0.53). One classic hallmark of bistability is that the distribution of dominance durations exhibit a long tail (occasional long durations), forming a gamma-like distribution (*Blake and Logothetis, 2002*). *Figure 2B* shows a clear long tail for dominance durations for the hallucinated bistability, consistent with the core characteristic of other bistable perceptual phenomena.

To obtain more objective performance based measures of this rotational bistability we tested whether sensitivity to retina-sourced physical motion stimuli presented within the annulus might differ when presented congruently or incongruently with the hallucinated motion. This would demonstrate an interaction between hallucinated content and physical discrimination – supporting a common mechanism. A new set of 10 participants tracked hallucinated motion alternations for

periods of 10 s. At a random time-point during the final 5 s of tracking a physical motion probe (*Figure 2C*; see Materials and methods) was presented in the annulus at either left or right of fixation and participants had to report on which side the probe was presented (a two alternative forced choice task). The probe was presented at one of six different contrasts (9.5%, 10%, 11%, 13%, 17%, 25%), set during pilot tests. Probe data was then separated based on probe direction, congruent or incongruent with the concurrent hallucination direction, and probe accuracy was fit with a non-linear function to give a threshold estimate of 70% accuracy (see Materials and methods). *Figure 2D* shows a significant difference between mean contrast thresholds for congruent and incongruent trials, with greater sensitivity to the probe stimulus when it was rotating in a congruent direction with the hallucinatory percept ($t(9) = 2.8$, $p=0.02$). This suggests that the hallucinatory motion percept (or tracking it) weakly suppresses or boosts *retina-based* motion depending on the motion congruity.

Next we wondered if hallucinated bistability, like perceptual retina-sourced bistability such as binocular rivalry, undergoes a local form of adaptation resulting in longer dominance durations over time (*Suzuki and Grabowecky, 2007*). For example, binocular rivalry alternations slow during viewing or when the stimulus is moved through visual space (*Blake et al., 2003*). Participants continuously tracked hallucinated alternations for a 5-min period in a small annulus, immediately followed by a further 5 min of tracking in a larger annulus, that did not spatially overlap with the prior stimulus (order was counter-balanced). *Figure 2E* shows that dominance durations increased over time during a session of same-sized stimulus. Further, in the second period of hallucination tracking, the new different-sized and non-overlapping annulus did not 'follow on' from the slower dynamics, but returned to the original shorter durations. These data suggest that hallucinated bistability can undergo a form of sensory adaptation that is local in visual space. To test if these changes in alternation rate were due to adaptation to the hallucinated content or due to adaptation to the actual perceptual flicker, six participants tracked alternations for 10 consecutive 30 s periods at three different flicker rates (8.5, 10.6 and 14.2 Hz). *Figure 2F* again shows adaptation, with an increase in dominance durations over time, however across the whole period the alternation rate was not statistically different between the three flicker rates ($F(2,24) = 1.76$, $p=0.19$), suggesting the change in alternation rate was not due to flicker adaptation, but most probably due to neural fatigue in neurons representing the hallucinatory bistable structure.

Another hallmark of perceptual bistability is the striking stabilization of the normal continuous stochastic dynamics by a sensory memory between intermittent presentations (*Pearson and Brascamp, 2008*). To learn if these bistable hallucinations show a similar sensory memory we presented the flickering annulus to eight participants for 2 s followed by 4 s without flicker (repeating intermittent presentation). Participants reported the dominant rotation direction on each 2 s presentation for 100 trials. *Figure 2G* shows the hallucination motion stability as the percentage of reported motion direction consecutively the same (e.g. clockwise, clockwise), such that reporting the same percept on every trial would result in 100% stability (*Pearson and Brascamp, 2008*). All individual subjects show a stability measure above the chance score of 50% (two rotation directions), with the mean significantly above chance (75%; compared to 50%: $p<0.001$), suggesting that hallucinated bistability can be stabilized by a novel form of memory across intermittent presentations.

Finally, to probe for potential overlapping mechanisms between hallucinated and perceptual or retina-sourced bistability 74 new participants tracked alternations in binocular rivalry, a bistable rotating sphere (see Materials and methods) and the flicker-induced hallucinations. *Figure 2H* shows a scatter plot of the data, the rotating sphere (red) significantly predicted hallucinated alternation rates ($r = 0.312$; $p=0.006$), while binocular rivalry (blue) did not ($r = 0.133$; $p=0.228$). We propose that the predictive relationship between the rotating sphere and hallucinated bistability, but not rivalry, might be due to the former two both involving a common neurophysiological mechanism for motion.

Next we extend a model developed for full field flicker hallucination to explain both the constrained hallucinated content and its bistability (*Rule et al., 2011*). This model is based on the idea that uniform luminance flicker stimulation resonates with the natural frequency of cortical cells to evoke standing waves of activity in primary visual cortex that induce the conscious experience of the hallucination (*Billock and Tsou, 2012*; *Rule et al., 2011*; *Ermentrout and Cowan, 1979*).

We modeled the region of visual cortex that was stimulated by the flickering annulus as a ring of tissue in one spatial dimension (*Figure 3A*). The neural tissue was modelled using an established neural field model (see Appendix 1: *Equations 1–3*) comprising spatially-coupled populations of

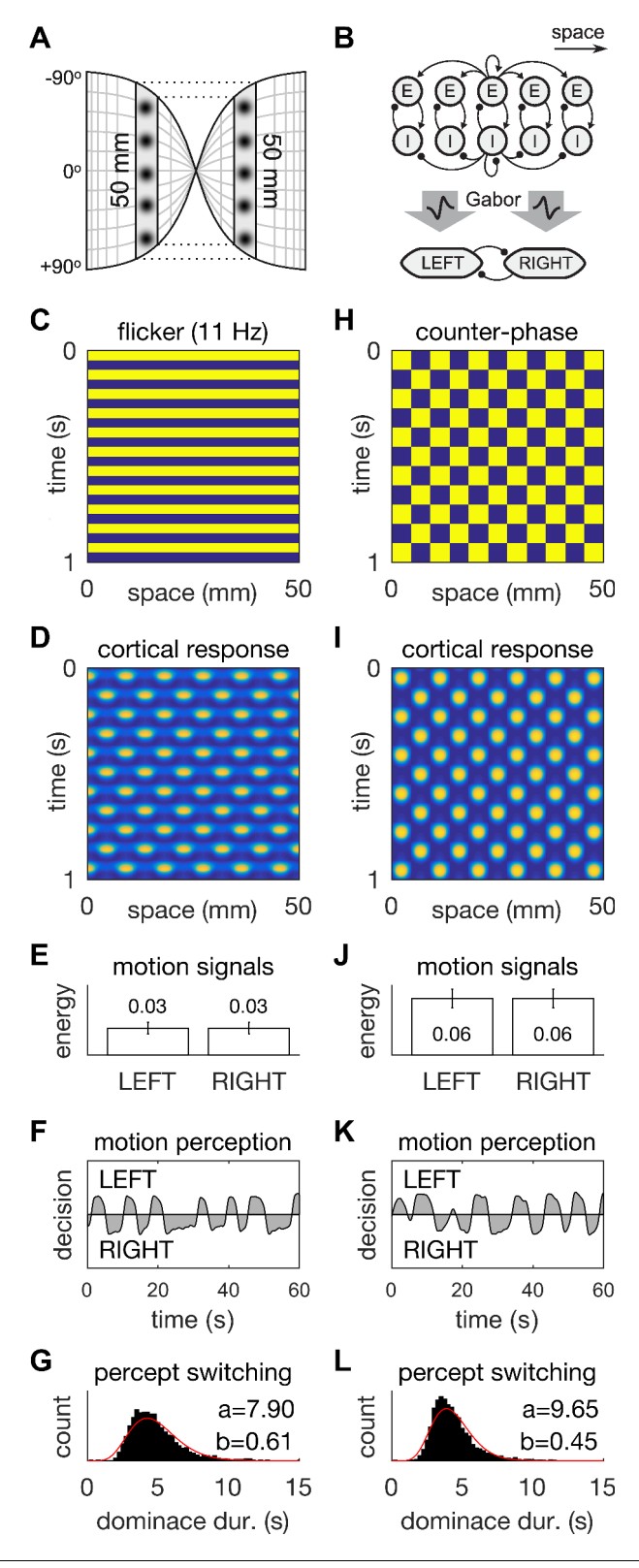

**Figure 3.** A neural field model of flicker-induced bistable motion. (**A**) The log-polar retinotopic map of human visual cortex. The fovea is located at the centre. The annulus stimulus maps onto a thin strip of tissue (shaded) that spans both hemispheres. Inter-hemispheric fibres (dotted lines) connect the strips to form a contiguous ring. (**B**) Schematic of the model. The ring of tissue is modelled in one spatial dimension using an established neural field

*Figure 3 continued on next page*

*Figure 3 continued*

model (***Rule et al., 2011***). that comprises local populations of excitatory (E) and inhibitory (I) cells. Flicker and counter-phase stimulation both induce counter-phase responses in this model. Motion within the cortical response patterns was detected by banks of Gabor filters (large arrows) following the motion-energy model. The motion-energy signals (boxes) represented the percepts of LEFT (anti-clockwise) and RIGHT (clockwise) motion. Those percepts were subject to rivalry through mutual inhibition and firing rate adaptation. (C) Space-time plots of flicker. (D) Cortical responses to flicker. (E) Time-averaged LEFT and RIGHT motion-energy responses to flicker stimulation. Error bars are ±1 standard deviation. (F) Time course of the perceptual decisions evoked by flicker. (G) Histogram of dominance durations for switching between left and right motion percepts fit with a gamma function. The variation in switch times is due to injected noise in perceptual rivalry model. (H–L) The analogous representations for the counter-phase retina-sourced stimulation.

The following figure supplement is available for figure 3:

**Figure supplement 1.** Motion percepts in the model under four different stimulus conditions.

---

excitatory (E) and inhibitory (I) cells (***Figure 3B***, top). The standing waves/hallucinations arise when flicker frequencies are approximately twice the natural frequency of the damped oscillations in the neural dynamics (***Rule et al., 2011***). The resonant frequency of simple cells in the primary visual cortex of cat typically peaks near 5 Hz (***Movshon et al., 1978***). Here we tuned the natural frequency of the model to 5.5 Hz so that it elicited prominent standing waves with 11 Hz flicker to match our behavioral data (***Figure 1D***).

Motion signals were then extracted from the space-time signatures of the standing waves in cortex using two banks of direction-selective motion detectors (***Figure 3B***, middle). These detectors (see Appendix 1: ***Equations 8–10***) were implemented using Gabor filters (***Jones and Palmer, 1987***) according to the classic motion-energy model (***Adelson and Bergen, 1985***). Given that our interest was in hallucinated motion, we applied the motion-energy model directly to the cortical activity patterns rather than to the stimulus pattern. The output of the motion detectors within each bank were linearly combined to form a net motion-energy signal that represented the valence of motion in the detector's preferred direction. These net motion signals were then fed into distinct neural populations that encoded the perceptual states of Left (counter-clockwise) and Right (clockwise) motion respectively (***Figure 3B***, bottom). Perceptual rivalry between these two neural populations was achieved through mutual inhibition and firing rate adaptation (see Appendix 1: ***Equations 11–13***). Models of this class replicate spontaneous perceptual switching between ambiguous stimuli and the association of higher switching rates with higher stimulus contrast/energy (***Laing and Chow, 2002***; ***Shpiro et al., 2007***; ***Wilson, 2003***). Independent noise was injected into the LEFT and RIGHT neural populations to induce variability in the dominance times of the rival motion percepts.

When stimulated with 11 Hz uniform flicker (the stimulus parameters that gave the highest hallucinated contrast; ***Figures 1D*** and ***3C***), the model generated 'cortical' standing waves, that is, oscillations in modelled cortex, at half the flicker rate (***Figure 3D***), in accordance with previous findings. Standing waves were observed for flicker frequencies in the range 8–18 Hz, which we interpret as hallucinatory. Further, it was also possible to induce spatially irregular patterns by adding in weak random connections between the E cells shown in ***Figure 3B*** (data not shown). Beyond those frequencies the cortical model produced spatially uniform responses (***Figure 3—figure supplement 1H***) which we interpreted as non-hallucinatory. The space-time signatures of the flicker-induced standing waves evoked identical responses from both the Left and Right motion detectors (***Figure 3E***). Consequently, those ambiguous motion signals induced spontaneous switching between the two populations (***Figure 3F***). The dominance times of each perceptual decision (***Figure 3G***) exhibited a long-tailed distribution ($M$ = 5.1 s, $SD$ = 1.6 s) that was qualitatively similar to those observed experimentally (***Figure 2B***). However, it is interesting to note the differences in the shape-parameter (a) of the gamma fits, between the behavioral and model data. This difference could be summed up by describing the model data as being closer to a normal distribution than the behavioral data. One possible reason for the difference could be our choice of Gaussian noise in the rivalry model. Future work could probe different noise distributions to fine tune a model of bistable hallucinations.

When the same model is presented with a counter-phase retina-sourced physical patterned stimulus (*Figure 3H*), with identical spatial (fx = 0.11 cycles/mm) and temporal (ft = 5.5 Hz) frequencies to the flicker-induced standing waves (hallucinations), standing waves were also produced in model-cortex (*Figure 3I*; see Appendix 2). The net motion signals were stronger than those induced by the blank flicker stimulation (*Figure 3E and J*), hence the spontaneous switching between LEFT and RIGHT perceptual decisions was somewhat faster (*M* = 4.3, *SD* = 1.5, *Figure 3K and L*).

## Discussion

Our data implicate retinotopically organized visual cortex as the site of bistable flicker induced hallucinations. Unlike previous work on visual hallucinations that suffered methodological limitations due to the almost infinite array of unpredictable hallucinated features (e.g. multiple combinations of color, form and motion), our technique essentially limits the space to a one dimensional annulus and the one set of visual features: moving light grey blobs.

The intriguing phenomenon of perceptual bistability has fascinated thinkers for centuries (*Blake and Logothetis, 2002*), both because of the phenomenological experience of oscillating consciousness, but also as it provides the informative dissociation between low-level stimulation and awareness. Here we show a hallucinated bistable stimulus, in which both the content (form and motion) and alternations in motion are endogenously generated. Counter-phase physical stimuli are known to induce motion percepts that spontaneously switch between the two candidate directions of motion. Our model suggests that uniform flicker can induce spatiotemporal responses in primary visual cortex that are very similar to those induced by counter-phase physical stimuli, given the appropriate choice of spatial and temporal frequencies. We argue that the same neural mechanism that contributes to apparent motion of a counter-phase physical stimulus also contributes to hallucinated experience. It is up to future research to divulge if sensory decisions in endogenously generated content are resolved using the same neural machinery as exogenously sourced perceptual information.

A neural field model based on the idea that standing waves of neural activity form visual hallucinations, provided a quantitative mechanism for these bistable hallucinations. It is interesting that in general, theories based on so-called symmetry-breaking standing waves have been proposed to explain the complex spatio-dynamics of the human brain (*Atasoy et al., 2016*), as well as in many other physical processes such as convection in fluids, animal coat markings, and cellular division (*Stewart, 1999*; *Turing, 1952*; *Kondo and Miura, 2010*).

The exceptional circumstances in which externally sourced stimuli are overwhelmed by internally generated spontaneous patterns of neural activity and the accompanying conscious experience (hallucinations), are notoriously difficult to study scientifically. Accordingly, almost no specific treatments have been developed for clinical use. Our technique for controlling and objectively measuring the range of hallucinated features (contrast, motion, bistability) and the corresponding neural model should prove useful in probing the mechanism(s) that allows such a range of non-ordinary function to produce hallucinations.

## Materials and methods

### Participants

Participants were students from the University of New South Wales who participated as part of a course requirement or were reimbursed for the time financially. No participants had a history of migraines, psychiatric or neurological disorders, and all had normal or corrected to normal eyesight. Informed written consent was obtained according to procedures approved by the ethics committee of the School of Psychology at the University of New South Wales.

### Experiment 1

Flicker frequency (1A: 8.5 Hz, 10.6 Hz, 14.2 Hz, 21.3 Hz; 1B: 4.7 Hz, 14.2 Hz, 21.3 Hz, 30.6 Hz): 65 participants completed experiment 1 (35 in experiment 1A, 24 female; 30 in experiment 1B, 18 female). Seven were excluded from experiment 1A and six from experiment 1B due to poor fitting

of a cumulative Gaussian function to their data (i.e. R2 <0.70). The final number of participants was 28 (21 female) and 24 (15 female) in experiments 1A and 1B, respectively.

### Experiment 2
Inter-ocular transfer: 29 first year psychology students (20 female) participated in experiment 2. Nine were excluded due to poor curve fitting (see above), leaving 20 participants (14 female).

### Experiment 3
Speed experiments (3A: variable flicker rate; 3B: variable eccentricity): six participants (one female) completed experiment 3A, and six (one female) completed experiment 3B. The author RC participated in both.

### Experiment 4
Gap experiments (4A: variable number of gaps; 4B: variable gap width): four individuals participated in experiment 4A (one female), and four participated in experiment 4B (one female), including the author RC in both.

### Experiment 5
Isoluminance experiment: 5 participants (one female) completed this experiment, including the author RC. Experiment 6. Bistable characteristics experiment: nine participants (three female) completed this experiment, including the author RC.

### Experiment 7
Probe experiment: 13 participants (10 female) completed this experiment. Data from three participants were excluded due to pre-set criteria indicating non-compliance with task instructions. The criteria were 1. failure to correctly detect probes on more than 25% of trials (chance score is 50%); and 2. monotonic functions of probe detection versus probe contrast with slopes less than 1.0 (including negative slopes), as performance should improve with higher probe salience.

### Experiment 8
Local adaptation experiments (8A: examination of local adaptation; 8B: functional impact of flicker rate): seven participants (one female) completed experiment 8A, and six (one female) completed experiment 8B, and the author RC participated in both experiments.

### Experiment 9
Intermitted presentation experiment: 8 participants (three female) completed this experiment, including the author RC.

### Experiment 10
Individual differences alternation rate experiment: 103 participants (67 female) completed this experiment. 19 were excluded due issues with reliability in their data (explained below). The total number of participants included in the analyses was 84 (57 female).

## Apparatus
Experiments 1, 2 and 7 were performed in a blackened room using a linearized CRT monitor at a resolution of 1600x1200 pixels and a refresh rate of 85 Hz. A chin rest was used to maintain a fixed viewing distance of 57 cm. Experiments 3, 4, 5, 6, 8, and 9 were performed on a different CRT monitor with a resolution of 1280x1024, a refresh rate of 85 Hz, and at a viewing distance of 57 cm. Experiment 10 was performed on a third CRT monitor with a resolution of 1280 x 960 pixels, a refresh rate of 85 Hz, and at a viewing distance of 47 cm.

## Stimuli

### Experiment 1

Flicker frequency (1A: 8.5 Hz, 10.6 Hz, 14.2 Hz, 21.3 Hz; 1B: 4.7 Hz, 14.2 Hz, 21.3 Hz, 30.6 Hz): The stimuli consisted of the flickering annulus and a smaller perceptual annulus composed of four sinusoidal luminance modulations arranged evenly about the annulus (*Figure 1B*). The inner and outer radii of the flickering annulus subtended 5.25° and 8.85° of visual angle, respectively, and its luminance during the on-frame was 74.97 cd/m$^2$. The inner and outer radii of the perceptual annulus were 3.5° and 1.4°, respectively, and the sinusoidal luminance modulation at the halfway point between the inner and outer radius was presented at 0.26 cycles per visual degree. The perceptual annulus was presented at one of the following eight Michelson contrasts on each trial: 4%, 16%, 32%, 40%, 48%, 64%, 80%, and 100%. The maximum luminance of the perceptual annulus (i.e. luminance at its brightest point) was the same as the flickering annulus during the on-frame (74.97 cd/m$^2$). The minimum luminance of each of these annuli, corresponding to the contrast values given above, were 61.98 cd/m$^2$, 53.33 cd/m$^2$, 33.77 cd/m$^2$, 23.55 cd/m$^2$, 17.22 cd/m$^2$, 7.24 cd/m$^2$, 1.94 cd/m$^2$, and 0.09 cd/m$^2$. The white fixation point had a radius of 0.150 and luminance of 74.97 cd/m$^2$. The luminance of the black background was 0.05 cd/m$^2$. In experiment 1A the flickering annulus was presented at 8.5 Hz, 10.6 Hz, 14.2 Hz and 21.3 Hz, and in experiment 1B at 4.7 Hz, 14.2 Hz, 21.3 Hz, and 30.6 Hz.

### Experiment 2

Inter-ocular transfer: Stimuli were shrunk so that they remained visible when viewed through a mirror stereoscope; the inner and outer radii of the flickering annulus subtended 2.7° and 5° of visual angle, and the inner and outer radii of the perceptual annulus subtended 0.75° and 2.15°. The eight contrast values of the perceptual annulus remained the same when the flickering annulus was presented at 21.3 Hz. However, results from Experiment 1 suggested that hallucinations evoked by 2.5 Hz flicker would be considerably lower in effective contrast. Thus, in blocks in which the annulus was presented at 2.5 Hz, perceptual annuli that would have been presented with a Michelson contrast of 100% were instead presented at 8% to ensure enough data points at lower contrasts for accurate curve fitting and interpolation. The luminance of the darkest section in this perceptual ring was 58.13 cd/m$^2$. The white fixation point had a radius of 0.140. In synchronous blocks, the annulus presented to each eye flickered in phase with the other, while in asynchronous blocks the annuli flickered exactly out of phase. The asynchronous blocks were presented at 2.5 Hz and then again at 21.3 Hz.

### Experiment 3

Speed experiments (3A: variable flicker rate; 3B: variable eccentricity): A single flickering white annulus was presented centrally on the screen, with an inner and outer radius of 7.08° and 8.83° in experiment 3A, respectively. In experiment 3B, the radii of the annulus varied between blocks such that the inner and outer radii, respectively, were as follows: small annulus, inner = 5.31, outer = 7.06; medium annulus, inner = 7.08°, outer = 8.83°; large annulus, inner = 8.79°, outer = 10.54°. In experiment 3A, the annulus flickered at 6 Hz, 8 Hz, and 10.6 Hz in separate blocks. In Experiment 3B, the annulus flickered at 10 Hz. In both experiments, the luminance of the white annulus and a concurrently-presented fixation point was 68.38 cd/m$^2$, and all stimuli were presented on a black background of luminance 0.21 cd/m$^2$. The fixation point subtended 0.30o of visual angle. In both experiments, two short green nonius lines (length and width both 0.45°) aligned along the midline were also presented, one situated above the annulus and the other below.

### Experiment 4

Gap experiments (4A: variable number of gaps; 4B: variable gap width): The stimuli consisted of a single flickering annulus located centrally on the screen (inner radius = 7.08°, outer radius = 8.83°), with a white central fixation point subtending 0.30° of visual angle. In both Experiment 4A and 4B, the annulus flickered at 14.2 Hz. In Experiment 4A, we added 4, 8, and 12 permanent gaps (each 0.59° wide) into the flickering annulus, angled along cardinal and sub-cardinal axes. Specifically, in the four-gaps condition, the annulus was divided into four equal-sized sectors along the cardinal direction; in the eight-gaps condition, the annulus was divided into eight equal-sized sectors along

the sub-cardinal lines; in the 12-gaps condition, it was further divided into 12 sectors. In Experiment 4B, we kept the number of spatial gaps at four and presented them only along cardinal axes. However, the width of gaps differed between blocks of trials (small: 0.59°, medium: 1.78°, large: 2.97°).

## Experiment 5

Isoluminance experiment: The stimuli consisted of a single flickering annulus located centrally on the screen (inner radius = 7.08°, outer radius = 8.83°), and a white central fixation point subtending 0.300 of visual angle. In different blocks of trials, we independently manipulated the flicker rate (6 Hz or 10.6 Hz) and the color and luminance of the annulus (a white annulus presented on a black background, or rapidly alternating isoluminant red and green annuli). The luminance parameters were 68.38 $cd/m^2$ and 0.21 $cd/m^2$ for white and black, respectively. For isoluminant red and green stimuli, the luminance parameters were determined separately for each participant by an isoluminance flicker fusion test, in which participants adjusted the brightness of a circle flickering between red (RGB triplet: 200, 0, 0) and green (RGB: 0, 150, 0) until the flicker was no longer perceptible (hence, a cessation of the perception of flicker).

## Experiment 6

Bistable characteristics experiment: The stimuli consisted of a single white annulus flickering at 15 Hz located centrally on the screen (inner radius = 7.08°, outer radius = 8.83°), and a white central fixation point subtending 0.30° of visual angle. The luminance of the black background was 0.21 $cd/m^2$, and 68.38 $cd/m^2$ for the annulus and fixation point.

## Experiment 7

Probe experiment. On each experimental trial a perceptual motion probe appeared abruptly and travelled within the boundaries of the flickering annulus stimulus. The motion probe appeared in one of two locations (left or right through the centre) within the annulus and travelled at a speed of 4.81° of visual angle per second for 500 ms through a quarter of the circumference of the annulus before abruptly disappearing again. The probe was drawn by frames of grey Gaussian blobs (diameter of 1.5°) that moved around the annulus. Probe contrast levels were denoted by reference to their darkest shade expressed as a percentage of the darkest possible on-screen shade such that 0% would be white (luminance 74.97 $cd/m^2$) and 100% would be black (luminance 0.5 $cd/m^2$). Using this formulation, the six contrast levels in order of ascending visibility were 9.5%, 10%, 11%, 13%, 17%, and 25%, and the corresponding luminance at the centre of each Gaussian blob were 54.09 $cd/m^2$, 52.43 $cd/m^2$, 51.45 $cd/m^2$, 49.02 $cd/m^2$, 43.31 $cd/m^2$, 33.45 $cd/m^2$ respectively.

## Experiment 8

Local adaptation experiments (8A: examination of local adaptation; 8B: functional impact of flicker rate): In experiment 8A, an annulus flickering at 14.2 Hz was presented centrally on the screen with a white central fixation point subtending 0.30o of visual angle, both with a luminance of 68.38 $cd/m^2$. The black background had a luminance of 0.21 $cd/m^2$. A small (inner radius = 5.65°, outer radius = 7.4°) and a large (inner radius = 9.080, outer radius = 10.830) annulus were presented for equal durations during each block. In experiment 8B, another flickering white annulus (inner radius = 7.080, outer radius = 8.830) with a white fixation point subtending 0.30°, both with luminance of 68.38 $cd/m^2$. The annulus in this experiment flickered at either 8.5, 10.6, or 14.2 Hz in separate 30 s periods. The luminance of the black background was 0.21 $cd/m^2$.

## Experiment 9

Intermitted presentation experiment: There were two alternating periods in this experiment: the on-period (2 s) and the off-period (4 s). In the on period, a single white annulus flickering at 14.2 Hz was presented centrally on the screen (inner radius = 7.08°, outer radius = 8.83°, luminance = 68.38 $cd/m^2$). In the OFF period, a non-flickering white annulus with the same dimensions and luminance was presented constantly for 4 s. A central fixation point subtending 0.30° of visual angle with luminance 68.38 $cd/m^2$ was presented throughout the on- and off-periods. Stimuli were presented on a black background with a luminance of 0.21 $cd/m^2$.

### Experiment 10

Individual differences alternation rate experiment. A white annulus flickering at 12.5 Hz was presented centrally on screen, with inner and outer radii of 7.24° and 10.88°, respectively, and a luminance of 77.73 cd/m$^2$. The binocular rivalry stimulus consisted of a green vertical sinusoidal grating presented to the right eye and a red horizontal grating presented to the left eye, which was achieved with red/green anaglyph glasses. The rotating sphere stimulus consisted of 300 white dots, each 0.14° in diameter, randomly distributed over the surface of a virtual sphere with a diameter of 6.33o. The sphere rotated rigidly about the vertical axis with a period of 1.25 s, giving the appearance of three-dimensional structure. A fixation point subtending 0.44° (luminance = 77.73 cd/m$^2$) was presented concurrently with the flickering annulus and binocular rivalry stimulus. All stimuli were presented on a black background with luminance 0.07 cd/m$^2$.

## Procedure

### Experiment 1

Flicker frequency (1A: 8.5 Hz, 10.6 Hz, 14.2 Hz, 21.3 Hz; 1B: 4.7 Hz, 14.2 Hz, 21.3 Hz, 30.6 Hz): On each trial, the flickering annulus, perceptual annulus, and the fixation point were presented concurrently and concentrically for eight seconds. Following this, a two-alternative forced choice question was displayed on screen, prompting participants to use the keyboard to indicate whether the perceptual annulus was lower or higher in contrast than the flickering annulus. Responses were not timed and trials progressed immediately upon receiving a response. The fixation point was replaced with a cross during the 2 s inter-trial interval. There was one block of 56 trials for each flicker frequency in each experiment, and block order was randomized between participants. The effective contrast of the hallucinations was estimated from a cumulative Gaussian curve fitted to the proportion of trials on which the participant reported the perceptual annulus as being lower in contrast than the flicker hallucinations at each contrast level of the perceptual annulus. This was done separately for each flicker frequency, using GraphPad Prism version 6.07 for Windows (Graph-Pad Software, San Diego, CA). A participant's data was excluded from subsequent analyses if any of the curve fitting attempts generated an R2 less than 0.7, as it was reasoned that the participant was unable to make the subjective judgements of hallucinations required for accurate contrast estimation.

### Experiment 2

Inter-ocular transfer: In this experiment, contrast was measured using the same method as described for study 1, however stimuli were presented for 7 s instead of 8. The fixation point remained on screen throughout the inter-trial interval, which lasted for 1.5 s. There were 56 trials per block and block order was randomized.

### Experiment 3

Speed experiments (3A: variable flicker rate; 3B: variable eccentricity): Participants were asked to attend to a specific patch of the hallucinatory pattern and monitor its movement in the annulus. Specifically, they focused on a greyish hallucinatory blob and tracked its trajectory moving in the right half of the annulus (marked by the nonius lines described above) while maintaining fixation at the central fixation point. Depending on the direction of hallucinatory motion (clockwise or counterclockwise), subjects were free to select either the upper or the lower nonius as the starting point of visual tracking, with the other nonius being the end point. They pressed and held a designated key to indicate the moment at which the monitored blob passed the starting point, and released the key when the blob arrived at the end point. If a change of direction occurred during the tracked travel time between the two nonius lines, participants pressed the spacebar and the trial was excluded from further analyses. In Experiment 3A there were three blocks of 40 trials (each 30 s in duration), one for each frequency (6, 9, and 12 Hz). In Experiment 3B, the stimulus flickered at a constant rate of 10 Hz. There were three different annulus sizes, presented in separate blocks, with 40 trials of 30 s in each condition.

## Experiment 4

Gap experiments (4A: variable number of gaps; 4B: variable gap width): In each trial, participants tracked the direction of their rotating hallucination in the flickering annulus for 60 s, by holding down one of two designated keys to indicate clockwise or counterclockwise direction, while ignoring the spatial gaps in the annulus. In Experiment 4A, there were three blocks of 10 trials, one block for each of the 4, 8, and 12 gaps conditions. In Experiment 4B, the number of gaps was held constant at four, and there was one block of 10 trials for each of the small, medium, and large gap conditions.

## Experiment 5

Isoluminance experiment: To equate the brightness of red and green stimuli, participants first completed the isoluminance test, in which they adjusted the brightness of a stimulus flickering between red and green until the cessation of perceptual of flicker. The brightness parameters were subsequently used for the red/green annulus in the main hallucination experiment. In each trial of the main experiment, participants viewed a flickering annulus (luminance: black/white; isoluminance: red/green; in separate blocks) and tracked their hallucinatory percepts for 30 s. They pressed one of three keys to indicate their percept: no hallucinatory pattern/shapes, only shapes, or both shapes and motion. The stimuli were flickered at 6 Hz or 12 Hz (in separate blocks). There were four blocks of 10 trials, giving 10 trials in each condition.

## Experiment 6

Bistable characteristics experiment: In this experiment, participants viewed a flickering annulus and tracked the rotating direction of hallucination continuously for 10 min, via holding one of the two designated keys down to indicate clockwise or counterclockwise motion.

## Experiment 7

Probe experiment: On each trial, participants were presented with the flickering annulus for 10 s. Throughout this period participants maintained their gaze on the fixation point and continuously tracked the rotation of the hallucination using their right hand to hold down one of three separate keys for clockwise, counter-clockwise, and ambiguous rotation. At a randomly set point during the second half of the 10 s flickering annulus presentation, a probe appeared within the annulus. Probe onset was restricted to the second half of the flickering annulus presentation to allow time for hallucinated rotation to commence. Hallucinated rotation at the moment of probe onset was identified not from the key being pressed at probe onset, but the key being pressed 400 ms after probe onset, to allow for participants' reaction time to press a key in response to a switch in perceived rotation direction. By way of this adjustment a perceptual switch immediately prior to probe onset would be accurately recorded. Post probe presentation, participants were prompted by on-screen text to report the location of the probe, using their left hand to press one of two keys to indicate whether the probe appeared on the left or right side of the annulus, thereby also indicating the rotation direction of the probe (probes on the left always moved counterclockwise, probes on the right always clockwise). In total there were eight blocks of 48 trials, with four trials per combination of probe contrast and location in each block.

## Experiment 8

Local adaptation experiments (8A: examination of local adaptation; 8B: functional impact of flicker rate): In Experiment 8A, participants viewed the flickering annulus and tracked the direction of the hallucination holding down one of two keys (for clockwise or counter- clockwise) continuously for 10 min. The annulus changed size (either from small to large or large to small) midway through the 10-min presentation (the change occurred at the onset of the 6th minute). All subjects completed two blocks; in one block, the smaller annulus was presented prior to the larger annulus, and the order was reversed in the second block. In each trial of experiment 8B, participants viewed the flickering annulus and tracked the hallucinatory motion for 30 s in the manner just described. There were three flicker rates in this experiment — 8 Hz, 12 Hz, and 16 Hz — presented in separate blocks of trials. Each 30 s tracking period was followed by an inter-trial interval of 5 s. Each block contained 10 trials.

## Experiment 9

Intermitted presentation experiment: In this experiment, the flickering annulus was presented in an intermittent fashion such that it was present for 2 s (on-period), followed by a static white ring for 4 s (off-period). Participants viewed the flickering annulus during the on-period and reported the dominant direction of perceived hallucinatory rotation by pressing a key to indicate clockwise or counterclockwise motion in the subsequent off-period. There was a single block of 100 trials; each 6 s trial was composed of a 2 s on-period and 4 s off-period.

## Experiment 10

Individual differences alternation rate experiment. In two separate two minute trials for each stimulus, participants tracked the direction of perceived motion of the hallucinatory blobs in the flickering annulus or rotating sphere, or the dominant pattern in binocular rivalry. Participants were required to hold down the key that best represented their perception at each moment in time. There were two keys for the two dominant perceptual alternatives of each stimulus (that is, one each for clockwise and counterclockwise rotation of flicker hallucinations, one each for rightward and leftward motion of the dots on the front face of the rotating sphere, and one each for the red horizontal and green vertical gratings in the binocular rivalry stimulus), and a third key for an ambiguous percept. Participants completed blocks for each stimulus type one after the other, but the order of pairs of blocks for each stimulus type was randomized between participants.

## Acknowledgements

We thank Selen Atasoy, Colin Clifford and Roger Keonig for helpful discussion and comments. JP is supported by Australian NHMRC grants GNT1046198, GNT1085404 and a Career Development Fellowship GNT1049596 and ARC discovery projects DP140101560 and DP160103299. SH and BE are supported by USA National Science Foundation award 1219753.

## Additional information

### Funding

| Funder | Grant reference number | Author |
|---|---|---|
| National Health and Medical Research Council | GNT1046198 | Joel Pearson |
| National Health and Medical Research Council | GNT1085404 | Joel Pearson |
| National Health and Medical Research Council | Career Development Fellowship - GNT1049596 | Joel Pearson |
| Australian Research Council | Discovery projects - DP140101560 | Joel Pearson |
| Australian Research Council | Discovery projects - DP160103299 | Joel Pearson |
| National Science Foundation | 1219753 | Stewart Heitmann Bard Ermentrout |

The funders had no role in study design, data collection and interpretation, or the decision to submit the work for publication.

### Author contributions

JP, Conception and design, Acquisition of data, Analysis and interpretation of data, Drafting or revising the article, Contributed unpublished essential data or reagents; RC, Conception and design, Acquisition of data, Analysis and interpretation of data, Drafting or revising the article; SR, MW, Acquisition of data, Analysis and interpretation of data, Drafting or revising the article; SH, Analysis and interpretation of data, Drafting or revising the article; BE, Analysis and interpretation of data, Drafting or revising the article, Contributed unpublished essential data or reagents

## Author ORCIDs

Joel Pearson, http://orcid.org/0000-0003-3704-5037

## Ethics

Human subjects: Participants were students from the University of New South Wales who partici-pated as part of a course requirement or were reimbursed for the time financially. No participants had a history of migraines, psychiatric or neurological disorders, and all had normal or corrected to normal eyesight. Informed written consent was obtained according to procedures approved by the ethics committee of the School of Psychology at the University of New South Wales.

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

## Appendix 1

# Numerical models

## Cortical model

The region of visual cortex that was stimulated by the flickering annulus was modelled by a ring of excitatory (E) and inhibitory (I) cells using an existing model of flicker-induced illusory patterns (**Rule et al., 2011**). Each cell in the model represents the average firing rate activity of a local population of neurons. Changes in firing rate followed the Wilson-Cowan equations (**Kilpatrick, 2013**; **Wilson and Cowan, 1972**, **Wilson and Cowan, 1973**)

$$\tau_e \dot{U}_e = -U_e + F(V_e + J) \tag{1}$$

$$\tau_i \dot{U}_i = -U_i + F(V_i + J) \tag{2}$$

where $U_e(x,t)$ and $U_i(x,t)$ represent the spatio-temporal activity of the excitatory and inhibitory cells respectively. Parameters $\tau_e$ and $\tau_i$ are the time constants of excitation and inhibition. The sigmoidal function,

$$F(V) = 1/(1 + \exp(-V)) \tag{3}$$

defines the firing rate of the cell where $V$ represents the net synaptic bombardment. Flicker stimulation was defined as $J(x,t) = H(\sin(2\pi f_t t))$ where $H$ is the Heaviside function. Whereas the standing wave stimulus was defined as $J(x,t) = H(\sin(2\pi f_t t) \sin(2\pi f_x x))$. The excitatory and inhibitory cells both received stimulation in equal amounts.

Cortical connection densities were assumed to be Gaussian with distance,

$$K(x) = \frac{1}{\sigma\sqrt{\pi}} \exp\left(\frac{-x^2}{\sigma^2}\right), \tag{4}$$

where $\sigma$ (mm) is the spatial spread. The net synaptic bombardment of each cell type was thus,

$$V_e = a_{ee} K_e \otimes U_e - a_{ei} K_i \otimes U_i - b_e \tag{5}$$

$$V_i = a_{ie} K_e \otimes U_e - a_{ii} K_i \otimes U_i - b_i \tag{6}$$

where the convolution operator,

$$K(x) \otimes U(x,t) = \int K(x-y) U(x,t) \, dy, \tag{7}$$

**Appendix 1—table 1.** Parameters of the cortical model.

| Parameter | Description |
| --- | --- |
| $U_e(x,t)$ | Excitatory activation state |
| $U_i(x,t)$ | Inhibitory activation state |
| $J(x,t)$ | Spatio-temporal stimulus |

*Appendix 1—table 1 continued on next page*

*Appendix 1—table 1 continued*

| Parameter | Description |
|---|---|
| $K(x)$ | Spatial coupling kernel |
| $\sigma_e = 0.8$ | Spread of excitation (mm) |
| $\sigma_i = 2$ | Spread of inhibition (mm) |
| $a_{ee} = 10$ | Coupling weight ($e$ to $e$) |
| $a_{ei} = 8.5$ | Coupling weight ($i$ to $e$) |
| $a_{ie} = 12$ | Coupling weight ($e$ to $i$) |
| $a_{ii} = 3$ | Coupling weight ($i$ to $i$) |
| $b_e = 2$ | Excitatory firing threshold |
| $b_i = 3$ | Inhibitory firing threshold |
| $\tau_e = 10$ | Excitatory time constant (ms) |
| $\tau_i = 30$ | Inhibitory time constant (ms) |
| $L = 100$ | Ring length (mm) |
| $dx = 0.1$ | Spatial resolution (mm) |
| $dt = 0.1$ | Integration time step (ms) |
| $f_x$ | Spatial frequency (cycles/mm) |
| $f_t$ | Temporal frequency (cycles/ms) |

represents the spatial summation of neural activity $U(x,t)$ according to the Gaussian coupling density $K(x)$. The coupling was also weighted by cell type, such that parameter $a_{ei}$ denotes the weight of the connection from cell type $i$ to cell type $e$. Parameters $b_e$ and $b_i$ are firing thresholds. See *Appendix 1—table 1* for parameter values.

## Motion-energy model

Motion detection in the cortical activity was simulated using the motion-energy model (*Anderson et al., 1991*). Specifically, the spatio-temporal response of a bank of motion detectors was defined as

$$M(x,t) = R_a^2(x,t) + R_b^2(x,t) \tag{8}$$

where

$$R(x,t) = \int G(x-y)\,U_e(x,t)\,dy. \tag{9}$$

defines the response of a simple receptive field with a preferred orientation and spatial wavelength. The squaring operation rectifies the response of the simple receptive field to approximate the frequency-doubled response a complex receptive field (*Hubel and Wiesel, 1962*). The simple receptive field is characterized by the two-dimensional Gabor function,

$$G(x,t) = \frac{1}{\sigma\sqrt{\pi}}\exp\left(\frac{-x^2}{2\sigma_x^2} + \frac{-t^2}{2\sigma_t^2}\right)\cos(2\pi f_x x + 2\pi f_t t - \phi), \tag{10}$$

where $\sigma_x = 4$ (mm), $\sigma_t = 30$ (ms), $f_x = 0.11$ (cycles/mm) and $f_t = 5.5$ (Hz). The phase offset $\phi$ (rad) determines the directional preference of the motion detector. Leftwards (anticlockwise) motion is detected with $\phi_a = 0$ and $\phi_b = \pi/2$ whereas rightwards (clockwise) motion is

detected with $\phi_a = 0$ and $\phi_b = -\pi/2$. The cortical activity was rescaled to $[-1, +1]$ prior to Gabor filtering by transforming $U_e \leftarrow 2U_e - 1$.

## Perceptual rivalry model

The net responses of the left and right motion detectors (**Passie et al., 2008**) were pooled into separate neural populations (labelled LEFT and RIGHT in **Figure 3B**) that competed for dominance via mutual inhibition. Spontaneous switching between opposing motion percepts was governed by slow spike rate adaptation (**Shpiro et al., 2007**). The activity in each neural population was governed by the dynamical equations

$$\tau_p \dot{P}_l = -P_l + F(-\gamma P_r - gZ_l + \bar{M}_l - \theta + \sigma \zeta(t)) \tag{11}$$

$$\tau_p \dot{P}_r = -P_r + F(-\gamma P_l - gZ_r + \bar{M}_r - \theta + \sigma \zeta(t)) \tag{12}$$

where $P_l(t)$ and $P_r(t)$ represent the firing rate of the LEFT and RIGHT neural populations, respectively. The two neural populations had identical time constants $\tau_p = 0.1$ (s) and firing thresholds $\theta = 0.2$. Both were also subject to independent Weiner noise $\zeta(t)$ with variance $\sigma = 0.002$. The strength of mutual inhibition was $\gamma = 0.7$ and the strength of adaptation was $g = 0.3$. The firing rate function $F(V) = 1/(1 + \exp(-kV))$ had slope parameter $k = 10$. The input to each population was the spatial average of its bank of motion detectors, namely $\bar{M}(t) = \int_0^L M(x,t)\,dx$.

Firing rate adaptation was governed by the dynamical equations

$$\tau_z \dot{Z}_l = P_l - Z_l + \sigma \zeta(t) \tag{13}$$

$$\tau_z \dot{Z}_r = P_r - Z_r + \sigma \zeta(t) \tag{14}$$

where $Z_l(t)$ and $Z_r(t)$ represent the adaptation of LEFT and RIGHT neural populations according to the slow time constant $\tau_z = 5$ (s). The adaptation variables were likewise subject to independent Weiner noise $\zeta(t)$ with variance $\sigma = 0.002$.

## Supplementary results

We also investigated the response of the model to non-illusory stimuli. Specifically we tested the model's response to true motion and to slow (2 Hz) uniform flicker stimulation. The true motion stimulus was a sinusoidal grating that moved smoothly leftwards at the same spatial frequency ($f_x = 0.11$ cycles/mm) and temporal frequency ($f_t = 11$ cycles/sec) as the counter-phase stimulus (panel B). It corresponds exactly to the preferred stimulus for the LEFT motion detector. The slow flicker stimulus is akin to a strobe stimulus with a frequency that is well below the range where hallucinations are observed in the psychophysics. The expectation was that neither the true motion stimulus nor slow strobe-like flicker should induce standing waves in the cortical model.

*Figure 3—figure supplement 1* compares the non-illusory responses with the illusory responses reproduced from the main text. The two leftmost columns represent the illusory responses and the two rightmost columns represent the non-illusory responses. To our surprise, the true motion stimulus (panel C) evoked standing waves in the cortical response (panel G) that were akin to those evoked by flicker and counter-phase stimulation (panels E and F). Nonetheless, the cortical response to true motion still contained more leftwards motion than rightwards motion. This is shown by the stronger response of the LEFT motion

detector ($\bar{M} = 0.062$) compared to that of the RIGHT detector ($\bar{M} = 0.034$) in panel K. That disparity in the two motion signals was sufficiently strong that the LEFT neural population consistently won the perceptual competition without any spontaneous switching (panel O). The perceptual decision to the true motion stimulus was thus steady and correct despite the formation of waves in cortex.

The strobe-like flicker stimulus (panel D) evoked no spatial patterning in the cortical response (panel H). This is consistent with previously published findings (*Rule et al., 2011*) where standing waves only arise for a limited band of flicker frequencies. Consequently, neither LEFT nor RIGHT motion signals were detected (panel L) and the perceptual decision (panel P) hovered around zero. Strobe-like flicker thus did not evoke any illusory sensation of motion, as is the case with the psychophysical observations.

