## [Decision Letter]

Thank you for submitting your article "Sensory dynamics of visual hallucinations in the normal population" for consideration by *eLife*. Your article has been reviewed by two peer reviewers, and the evaluation has been overseen by a Reviewing Editor and David Van Essen as the Senior Editor. The following individuals involved in review of your submission have agreed to reveal their identity: Alex Meier (Reviewer #1); Hugh Wilson (Reviewer #2).

The reviewers have discussed the reviews with one another and the Reviewing Editor has drafted this decision to help you prepare a revised submission.

Summary:

The paper describes a new form of flicker-induced illusion involving moving grey blobs in an annulus which increased with flicker rate, was eccentricity independent, was subject to local sensory adaptation and was bistable, displaying some of the features of low-level visual phenomena. The authors refer to the illusion as a form of a hallucination and suggest that the tools used to characterize the illusion could be used to characterize hallucinations. Based on the data and the model developed to account for the observed illusion the authors postulate the same neural mechanism that underlies the apparent motion induced by counterphase flicker, also underlies the reported flicker induced bistable illusory motion. Both reviewers were very positive about the work, complimenting the experimental design, the analysis and clear writing. They raised only a few relatively minor issues that the authors should address in the revision. These are listed below:

Essential revisions:

1) Results section (fifth paragraph). Please respond to the Reviewer 1 point that "binocular neurons" should not be equated with "cortical neurons", as binocular responses are also present in the LGN and in the superior colliculus.

2) Please provide a disclaimer to the interpretation of the isoluminance experiment that the b/y channel has not been tested. That point may be relevant because binocular LGN responses have been reported to be exclusive to koniocellular neurons that also are selective for s-cone stimuli.

3) For each graph provide information about the sample number and dimension (i.e., n=x subjects or n=y trials).

4) Reviewer 2 pointed out that unlike the spatially patterned Video #2, gray blob-like patches in Video #1, did not appear to be spatially periodic. He asked whether "such irregular blobs were the percept in the experiments, or is the irregularity an artifact of the video generation process? If blob spatial irregularity was perceived in the experiments, then does the model also explain this (presumably with noise in spatial connectivity strengths it could)? ". Please comment.

5) Abscissa in Figure 2 should be in seconds rather than number of bins. Also, show a γ function that fits these data. For direct comparison, γ functions should be fit to the model results in Figure 3. The parameters for the γ function fits should be reported.

Reviewer #1:

This is an interesting, well done and timely study on a fascinating new perceptual phenomenon. The individual experiments that make up the paper are all well designed and analyzed with care. The outcomes are communicated clearly and put into an appealing wider context. There are no major issues that need to be addressed before publication.

Reviewer #2:

This is an interesting and important paper that describes a new form of flicker-induced illusion of moving gray blobs in an annulus. The authors have made an extensive set of measurements of many aspects of this illusion, including rivalry between motion directions. A detailed quantitative neural model of cortical processing is then shown to account for the data. Publication is strongly recommended after the two points below are resolved.

While viewing Video #1, I definitely perceived gray blob-like patches that did seem to move. However, these were not periodic spatially, as in the spatially patterned Video #2. Were such irregular blobs the percept in the experiments, or is the irregularity an artifact of the movie generation process? If blob spatial irregularity was perceived in the experiments, then does the model also explain this (presumably with noise in spatial connectivity strengths it could)?

In Figure 2, it is appropriate to have the abscissa labeled in seconds rather than number of bins. Also, a γ function fit to these data should be shown. For direct comparison to this, γ functions should be fit to the model results in Figure and 3L. Obviously, parameters for the γ function fits should be reported.

---

## [Author Response]

Essential revisions:

1) Results section (fifth paragraph). Please respond to the Reviewer 1 point that "binocular neurons" should not be equated with "cortical neurons", as binocular responses are also present in the LGN and in the superior colliculus.

We thank the reviewer for pointing this out and have changed the phasing to be more accurate in the Results section.

2) Please provide a disclaimer to the interpretation of the isoluminance experiment that the b/y channel has not been tested. That point may be relevant because binocular LGN responses have been reported to be exclusive to koniocellular neurons that also are selective for s-cone stimuli.

Good point, we have added the disclaimer to the seventh paragraph of the Results section of the manuscript.

3) For each graph provide information about the sample number and dimension (i.e., n=x subjects or n=y trials).

These have been added to the figure legends for the empirical figures.

4) Reviewer 2 pointed out that unlike the spatially patterned Video #2, gray blob-like patches in Video #1, did not appear to be spatially periodic. He asked whether "such irregular blobs were the percept in the experiments, or is the irregularity an artifact of the video generation process? If blob spatial irregularity was perceived in the experiments, then does the model also explain this (presumably with noise in spatial connectivity strengths it could)? ". Please comment.

Great observation. Yes, people often report irregular blobs under controlled viewing conditions. The model can indeed produce irregular blobs, by simply adding weak random connections between the E cells of the model, shown in Figure 3.

Author response image 1.**DOI:**
http://dx.doi.org/10.7554/eLife.17072.010

Figure 4 shows some new output from the model showing irregular blobs, time is shown on the vertical axis and space on the horizontal. We have added text to the manuscript to explain that this is possible (Results, sixteenth paragraph).

5) Abscissa in Figure 2 should be in seconds rather than number of bins. Also, show a γ function that fits these data. For direct comparison, γ functions should be fit to the model results in Figure 3. The parameters for the γ function fits should be reported.

Good point, we changed the Abscissa to seconds and have fit a γ function to these data, shown in the new Figure 2. The fits are reported in the figure legend. Likewise, we have fit a γ function to the model data, shown in the new Figure 3.